# Unveiling new quantum phases in the Shastry-Sutherland compound SrCu₂(BO₃)₂ up to the saturation magnetic field

T. Nomura [1,2] ✉, P. Corboz [3] ✉, A. Miyata[4], S. Zherlitsyn [4], Y. Ishii [1], Y. Kohama [1], Y. H. Matsuda [1], A. Ikeda [5], C. Zhong [6,8], H. Kageyama [6] & F. Mila [7]

Under magnetic fields, quantum magnets often undergo exotic phase transitions with various kinds of order. The discovery of a sequence of fractional magnetization plateaus in the Shastry-Sutherland compound SrCu₃(BO₃)₂ has played a central role in the high-field research on quantum materials, but so far this system could only be probed up to half the saturation value of the magnetization. Here, we report the first experimental and theoretical investigation of this compound up to the saturation magnetic field of 140 T and beyond. Using ultrasound and magnetostriction techniques combined with extensive tensor-network calculations (iPEPS), several spin-supersolid phases are revealed between the 1/2 plateau and saturation (1/1 plateau). Quite remarkably, the sound velocity of the 1/2 plateau exhibits a drastic decrease of -50%, related to the tetragonal-to-orthorhombic instability of the checkerboard-type magnon crystal. The unveiled nature of this paradigmatic quantum system is a new milestone for exploring exotic quantum states of matter emerging in extreme conditions.

Thanks to its sequence of magnetization plateau phases at fractional magnetization, SrCu₂(BO₃)₂ (SCBO) is one of the most celebrated frustrated magnets[1–4]. In this material, Cu²⁺ ions with spin $S = 1/2$ arrange in an orthogonal-dimer geometry known as the Shastry-Sutherland lattice (Fig. 1a, b)[5,6]. The effective Hamiltonian including intra- and inter-dimer interactions ($J$ and $J'$, respectively) and the Zeeman term is defined by

$$\mathcal{H} = J \sum_{\langle i,j \rangle} S_i S_j + J' \sum_{\langle\langle i,j \rangle\rangle} S_i S_j - h \sum_i S_i^z. \qquad (1)$$

The competing antiferromagnetic (AFM) couplings ($J$ and $J'$) result in a geometrical frustration, giving rise to various phases depending on the ratio $J'/J$ and magnetic field $h$[7–14]. The exchange parameter ratio for SCBO at ambient pressure is estimated to be $J'/J \sim 0.63$ from the magnetization measurement up to 118 T[3]. However, the pantograph-like magnetostriction, which modulates the Cu-O-Cu angle and, hence, the superexchange interaction, can change the ratio in applied magnetic fields[15,16]. At low temperatures, SCBO shows a fascinating magnetization curve with multiple anomalies, the most prominent ones being related to the plateau phases at 1/8, 2/15, 1/6, 1/4, 1/3, 2/5, and 1/2 of the saturation magnetization $M_s$[3,17–19]. In the plateau regions, triplets (or bound states of triplets for the low-field plateaus) crystallize into magnetic superstructures[20,21] as a result of the effective repulsion and of the localized nature of triplets inherent to the frustrated

[1]Institute for Solid State Physics, University of Tokyo, Kashiwa, Chiba, Japan. [2]Tokyo Denki University, Adachi, Tokyo, Japan. [3]Institute for Theoretical Physics and Delta Institute for Theoretical Physics, University of Amsterdam, XH Amsterdam, The Netherlands. [4]Hochfeld-Magnetlabor Dresden (HLD-EMFL), Helmholtz-Zentrum Dresden-Rossendorf, Dresden, Germany. [5]Department of Engineering Science, University of Electro-Communications, Chofu, Tokyo, Japan. [6]Graduate School of Engineering, Kyoto University, Nishikyouku, Kyoto, Japan. [7]Institute of Theoretical Physics, Ecole Polytechnique Fédérale de Lausanne (EPFL), Lausanne, Switzerland. [8]Present address: Department of Applied Chemistry, Ritsumeikan University, Kusatsu, Shiga, Japan. ✉e-mail: tnomura@mail.dendai.ac.jp; P.R.Corboz@uva.nl

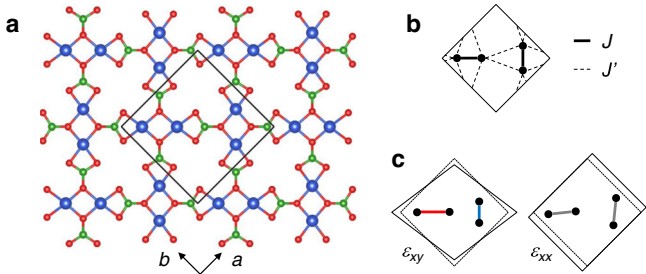

**Fig. 1 | Crystal structure of SrCu₂(BO₃)₂ and Shastry-Sutherland model. a** Crystal structure in the *ab* plane. The blue, red, and green spheres represent Cu, O, and B atoms, respectively. The black square shows the crystallographic unit cell. **b** Exchange couplings ($J$ and $J'$) between Cu²⁺ ions with $S = 1/2$. **c** Deformed structure with the strains $\varepsilon_{xy}$ ($c_{66}$ mode) and $\varepsilon_{xx}$ ($c_{11}$ mode), resulting in the asymmetric and symmetric modulations of $J$, respectively.

geometry[6,22–25]. So far, no experimental investigation up to the saturation magnetization has been reported for this material.

In this paper, we present the results of ultrasound and magnetostriction measurements up to 150 T, reaching for the first time the saturation field. For studying the magnetism of SCBO at extremely high magnetic fields, the ultrasound and magnetostriction are powerful techniques due to presence of the strong spin-strain interactions. Furthermore, the sensitivity of these techniques is maximal at the top of the pulsed field, where the sensitivity is strongly reduced for the magnetization measurements[26]. Besides, the earlier studies have pointed out that the spin-lattice coupling plays an important role for the high-field properties of SCBO[15–17,27–30]. In this work, we investigate the magneto-structural properties of SCBO at ultrahigh magnetic fields supported by theoretical calculations based on the infinite projected entangled pair state (iPEPS) tensor-network algorithm[31–33]. The elastic properties of novel supersolid phases above 100 T are discussed in terms of the spin-lattice coupling.

## Results

### Ultrahigh-field results

Figure 2 shows a summary of the ultrasound and magnetostriction results up to the ultrahigh field of 150 T. For comparison, the magnetization and its field-derivative curves at 2.1 K[3] are shown in Fig. 2c with bars representing the regions of magnetization plateaus. For reproducibility and raw data, see Supplementary Information (SI).

First, we discuss the results of the sound velocity. Figure 2a shows the relative changes of the sound velocity $\Delta v/v_0$ for the $c_{11}$ mode (multiplied by 10) and the $c_{66}$ mode. The results obtained by using the non-destructive magnet are shown by the black curves. $\Delta v/v_0$ is significantly larger for the $c_{66}$ than for the $c_{11}$ mode, which is consistent with the previous study[28]. Clear anomalies are observed at 27, 33, 40, and 74 T, corresponding to the onsets of the 1/8, 1/4, 1/3, and 2/5 plateau phases, in line with published high-field results[3,17]. The results above the 1/2 plateau are obtained by the single-turn coil (STC) experiments (colored curves). Therefore, the experimental error (±10% of $\Delta v/v_0$) is larger than that estimated from the non-destructive pulsed-magnet experiments (±3% of $\Delta v/v_0$). Nevertheless, the relative changes in $\Delta v/v_0$ are reliable and qualitatively well reproduced. The sound velocity of the $c_{66}$ mode stays constant at $\Delta v/v_0 = -50\%$ in the 1/2 plateau phase. This is a surprising result because such a large softening is usually observed at the phase boundary when a soft mode leads to a lattice distortion[30]. The strong softening persists up to 116 T, which is slightly higher than the end of the 1/2 plateau (108 T) reported by the magnetization measurement[3]. At 116 T, $\Delta v/v_0$ shows a drastic increase. Another anomaly is observed at 126 T, where the slope of $\Delta v/v_0$ changes. One more anomaly is detected at 140 T, where $\Delta v/v_0$ discontinuously increases and saturates at the level of + 20%. The slight difference between the field up- and down-sweeps might be due to the

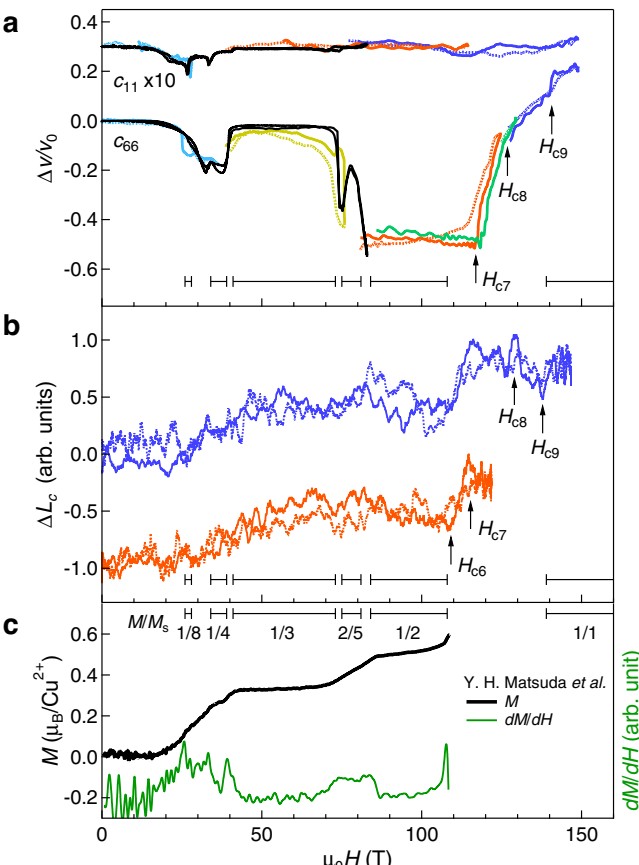

**Fig. 2 | The ultrahigh-field data ($H//c$) obtained in SrCu₂(BO₃)₂. a** Relative change of the sound velocity for the $c_{66}$ and $c_{11}$ acoustic modes. The results for the $c_{11}$ mode are multiplied by 10. The results obtained with the non-destructive magnet at 1.5 K and the STCs at 3.2 K are shown by the black and colored curves, respectively. The purple, green, orange, yellow, cyan curves represent the results up to the maximum fields of ⁓150, 130, 120, 70, 30 T, respectively. **b** Magnetostriction along the *c* axis measured at 4.2 K. The results up to the fields of 122 and 146 T are shown by orange and purple, respectively. **c** Magnetization (left) and its field derivative (right) measured at 2.1 K[3]. The plateau field regions suggested by the magnetization measurement are shown in bars. Anomalies above the 1/2 plateau are denoted by arrows. The results in the field up- and down-sweeps are shown by the solid and dotted lines, respectively.

temperature change and/or the lattice dynamics at the magneto-structural transitions. In contrast to the $c_{66}$ mode, the $c_{11}$ mode reveals no clear anomaly above 80 T within our experimental resolution.

Figure 2b shows two magnetostriction curves measured along the *c* axis. The results exhibit features similar to those of the magnetization; both start to increase when the spin gap closes (⁓25 T) and stay approximately constant in the plateau phases. Here, we comment on the reproducibility of our magnetostriction results. As the previous study shows[17], the magnetostriction strongly depends on the experimental setting and the strain between the fiber-Bragg grating (FBG) and the sample. Indeed, we also find that the results vary slightly depending on the glue and the surrounding grease. Although the overall magnetostriction depends on the setting, the transition field detected by this technique is well reproduced. Therefore, in this study, we only focus on the critical fields and do not discuss the magnitude of the magnetostriction. Above the 1/2 plateau, we detect four anomalies in $\Delta L_c$ at 108, 116, 128, and 138 T.

Table 1 summarizes the critical fields obtained by the ultrasound, magnetostriction, and magnetization experiments. The critical-field notations ($H_{c6}, H_{c7}, H_{c8}$, and $H_{c9}$) are termed after Ref. 3. The critical-field values agree well considering the error range of the STC

experiments. The only discrepancy is that the anomaly at 108 T is missing in the ultrasound results. This anomaly corresponding to the end of the 1/2 plateau is observed in both magnetization and magnetostriction. Note that the experimental conditions (sample cooling and sweep rate) are very similar for these three experiments. Therefore, it suggests that the acoustic modes ($c_{11}$ and $c_{66}$) are not sensitive to the transition at 108 T. The origin of the observed anomalies is discussed below.

### Theoretical phase diagram

In Fig. 3, we present the phase diagram as a function of $J'/J$ between the 1/2 plateau and full saturation, for values of $J'/J$ in the vicinity of the predicted value $J'/J = 0.63$ for SCBO from Ref. 3. The data have been obtained for a bond dimension $D = 8$ and cluster update optimization, which is sufficiently large so that finite-$D$ errors on the phase boundaries are small (of the order of the symbol sizes). Representative spin patterns of the phases are also shown in Fig. 3. Besides the familiar 1/2 plateau phase, we find different types of supersolid phases (SSPs), i.e., phases that simultaneously break translational symmetry and the U(1) symmetry associated with the total $S^z$ conservation. They all exhibit a diagonal stripe pattern with a certain period.

Above the 1/2 plateau the dominant phase is the 1/3 SSP, which has a period 6 and which can also be found at lower fields above the 1/3 plateau (hence the name 1/3 SSP, see Ref. 3). Within the 1/3 SSP, there are two distinct regions which we call type a and type b. The former has 2 different spin directions, whereas the latter has 6. In between the 1/2 plateau and the 1/3 SSP for $J'/J \leq 0.63$, an extremely narrow period-10 SSP appears, which is energetically very close to their neighboring phases. We confirmed that the extent of the period-10 SSP further increases with decreasing $J'/J$ from additional simulations down to $J'/J = 0.56$ (see SI). At higher magnetic fields we find a transition into a period-14 SSP before reaching saturation. Near the saturation, the energies of the competing states get very close. At larger values of $J'/J \sim 0.68$ a period-8 SSP is stabilized before saturation. We did not find evidence of another plateau above the 1/2 plateau, although a 7/8 plateau gets energetically close to the supersolid phases, especially at larger values of $J'/J$.

In Fig. 4, the magnetization curve for $J'/J = 0.63$ together with the energy differences of the competing states is shown, where the vertical dashed lines indicate the phase boundaries. Converted to real units[3], we find that the 1/2 plateau terminates at 106 T, with a jump in magnetization to the extremely narrow period-10 SSP, followed by the 1/3 SSP. The location of this transition is compatible with $H_{c6}$ from the magnetostriction and magnetization measurements. The transition between the two 1/3 SSPs is found at 122 T (there is no anomaly observed in experiments at this value which could be because the two 1/3 SSPs are very similar states). The transition into the period-14 SSP occurs around 128 T which coincides with $H_{c8}$. Finally, saturation is reached at 137 T, in close agreement with the experimental values of $H_{c9}$.

## Discussion

### Sound velocity of the plateau phases

First, we quantitatively discuss the drastic sound-velocity change of the $c_{66}$ mode. The sound velocity $v$ is a thermodynamical quantity, related to the elastic constant $c$, as $c = \rho v^2$. Here, $\rho$ is the mass density. The elastic constant is the second derivative of the free energy with respect to the strain. In SCBO, strain modulates the Cu-O-Cu angle and exchange coupling of the dimers, leading to a modulation of the magnetic free energy. Depending on the elastic modes ($c_{66}$ and $c_{11}$), the exchange modulation acts on dimers asymmetrically and symmetrically, respectively (Fig. 1c). For the case of the $c_{66}$ mode with $\varepsilon_{xy}$, the Cu-O-Cu angle of the horizontal (vertical) dimer decreases (increases), leading to the reduced (enhanced) AFM interaction. This bond alternation naturally stabilizes the 1/2 plateau phase with the checkerboard pattern; i.e., vertical dimers with (almost) aligned spins and horizontal dimers with predominantly singlets (or vice versa). This is in strong contrast to the 1/3 plateau which exhibits polarized spins on both the vertical and horizontal dimers (separated by two rows with dimers with opposite spins on each dimer[3]). This odd periodicity leads to the cancellation of contributions when computing the derivative, leading to a value close to zero. The 1/4 plateau exhibits an even periodicity (every fourth diagonal row exhibits almost polarized spins), leading to a finite first derivative, albeit with a smaller magnitude than the 1/2 plateau, as observed also in the experiment. At saturation, the state is a product state

**Table 1 | Critical fields obtained by the ultrasound, magnetostriction, and magnetization experiments**

|  | $H_{c6}$ | $H_{c7}$ | $H_{c8}$ | $H_{c9}$ |
|---|---|---|---|---|
| Ultrasound | — | 116(3) | 126(3) | 140(2) |
| Magnetostriction | 108(2) | 116(2) | 128(3) | 138(3) |
| Magnetization[3] | 108(1) |  |  |  |

The field values with error ranges are shown in the unit of Tesla.

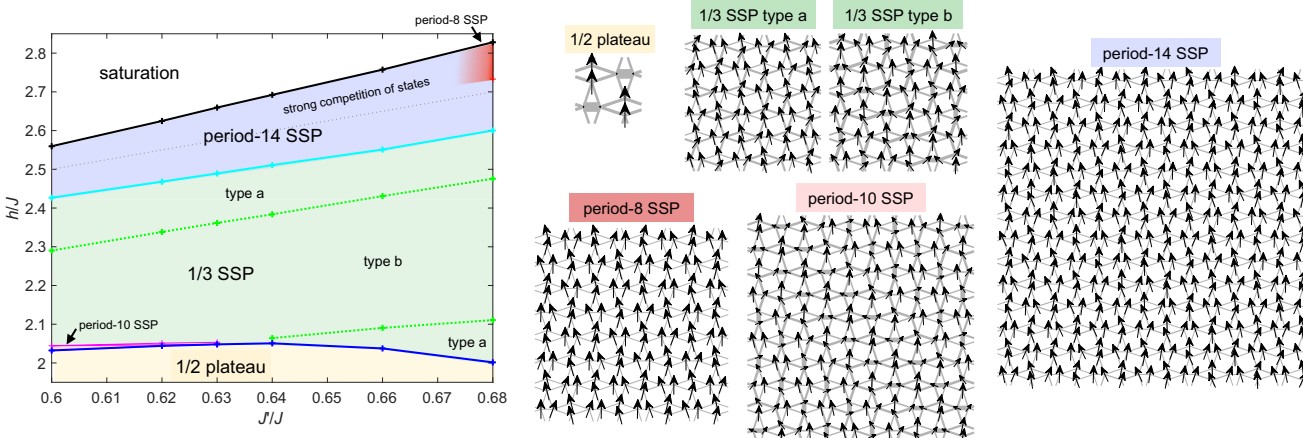

**Fig. 3 | iPEPS phase diagram of the Shastry-Sutherland model at high magnetic fields up to saturation.** The results are obtained for $D = 8$ and cluster optimization, revealing several supersolid phases (SSPs) with different periods in between the 1/2 plateau and saturation. Typical spin patterns for each phase are shown, where the thickness of the grey bonds scales with the local bond energy (the thicker the lower the energy).

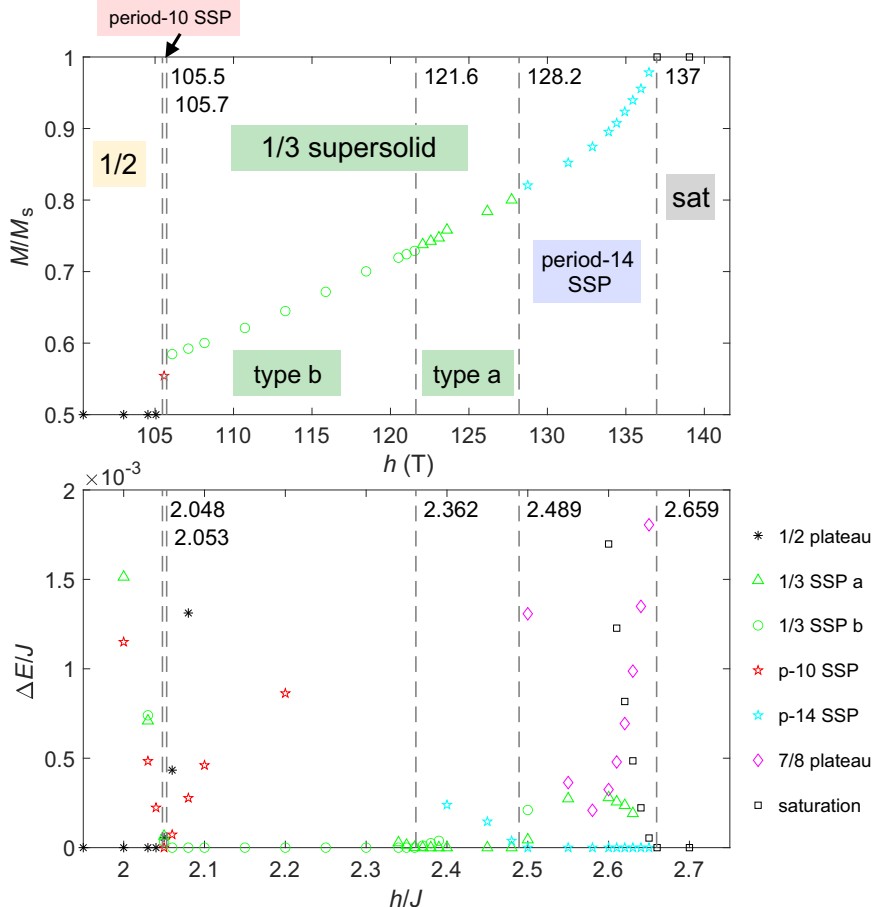

**Fig. 4 | Energy landscape of the Shastry-Sutherland model near the saturation magnetic fields.** Upper panel: Magnetization curve obtained with iPEPS for $J'/J = 0.63$ ($D = 8$ and cluster update optimization) at high fields up to saturation. Lower panel: Energy difference in units of $J$ from the lowest energy state. The values on the horizontal axis in the upper panel have been converted to real units[3] for direct comparison with experiments. Phase boundaries are indicated by vertical dashed lines.

(all spins aligned) and the derivatives with opposite signs cancel each other exactly.

Table 2 summarizes the first and second derivatives of the free energy ($E'$ and $E''$) for the $c_{66}$ mode. Here, we have evaluated the energy shift with respect to $\varepsilon_{xy}$, which reduces (enhances) the AFM exchange coupling of the horizontal (vertical) dimer. As discussed, $E'$ is much larger in the 1/2 plateau than in other plateau phases. This is a common feature with the experimental $\Delta v/v_0$, where the sound velocity significantly decreases in the 1/2 plateau but does not in the 1/3 plateau. In fact, the experimental $\Delta v/v_0$ (and $\Delta c/c_0$) of the major plateau phases at 1/4, 1/3, and 1/2 is nearly proportional to $E'$ (see the second, fifth, and seventh columns of Table 2), suggesting that $E'$ gives a major contribution to the elastic constant as compared to $E''$ although the elastic constant is the second derivative of the free energy. In fact, $E'$

contributes to the elastic constant via the anharmonic potential of the lattice (see SI for details). This anharmonicity is specifically important for the 1/2 plateau phase because the triplet crystallization into the checkerboard pattern leads to the cooperative exchange striction from tetragonal to orthorhombic. Because of this exchange striction, $Cu^{2+}$ ions move from the equilibrium positions at zero field to strained positions where the magnetic superstructure is better stabilized. At the strained position, the effect of the anharmonic potential becomes relevant, which is a key to connect $E'$ and the elastic constant. Including the first- and second-derivatives contributions as $\lambda_1 E' + \lambda_2 E''$, the experimental $\Delta v/v_0$ is reasonably explained (see the sixth and seventh columns of Table 2). The coefficients $\lambda_1$ and $\lambda_2$ are optimized by a least-squares fit.

## High-field supersolid phases
The iPEPS calculation predicts four SSPs between the 1/2 plateau and the saturation. Even if $J'/J$ changes under high fields because of the exchange striction, the phase boundary does not change greatly. In our experiments, the period-10 SSP and the transition from the 1/3 plateau type b to type a would not be detected because of the limited resolution and precision. Thus, we focus on the major phase boundaries, the end of the 1/2 plateau, the end of the 1/3 SSP, and the saturation field. The latter two are detected in both ultrasound and magnetostriction experiments as $H_{c8}$ and $H_{c9}$. The quantitative agreement between the experiment and theory is excellent considering the experimental challenges at ultrahigh magnetic fields. The slight

## Table 2 | Experimental and theoretical sound velocity

| $M/M_s$ | $E'$ | $E''$ | $\Delta c/c_0$ iPEPS | $\Delta c/c_0$ Exp. | $\Delta v/v_0$ iPEPS | $\Delta v/v_0$ Exp. |
|---------|------|-------|----------------------|---------------------|----------------------|---------------------|
| 1/8 | −0.014 | −0.44 | −0.24 | −0.21(8) | −0.13 | −0.11(4) |
| 1/4 | −0.051 | −0.31 | −0.29 | −0.34(5) | −0.16 | −0.19(3) |
| 1/3 | 0 | −0.18 | −0.08 | −0.06(4) | −0.04 | −0.03(2) |
| 1/2 | −0.23 | −0.034 | −0.74 | −0.73(11) | −0.49 | −0.48(7) |

First and second derivatives of the energy with respect to the strain $\varepsilon_{xy}$ ($c_{66}$ mode) obtained with iPEPS ($D = 6$) are summarized. The iPEPS estimates of $\Delta c/c_0$ in the fourth column have been obtained based on a least-squares fit of the form $\lambda_1 E' + \lambda_2 E''$ to the experimental values of $\Delta c/c_0$.

difference in the saturation field might be due to the magnetic-field dependence of $J$.

The assignments of the lower-field boundaries $H_{c6}$ and $H_{c7}$ require careful discussions. The transition at $H_{c6}$ is the end of the 1/2 plateau, where magnetization starts to increase discontinuously. This anomaly is observed in the magnetostriction and magnetization experiments but is not detected in the ultrasound experiment. Another transition at $H_{c7} = 116$ T is observed both in ultrasound and magnetostriction experiments but is not predicted by the iPEPS calculation. Experimentally, the $\Delta v/v_0$ of the $c_{66}$ mode shows a drastic increase at $H_{c7}$ (Fig. 2a).

From the experimental point of view, the transition from the 1/2 plateau to the 1/3 SSP, where the magnetic unit cell drastically reconstructs, should be detected by the ultrasound technique. The ultrasound technique is generally sensitive to this kind of symmetry change at phase transitions[30]. Indeed, the calculated $E'$ is significantly larger in the 1/2 plateau because of the checkerboard pattern of triplets with the even period, while it is almost canceled with the odd period of the 1/3 SSP. The ultrasound results without anomaly at $H_{c6}$ suggest that another SSP with an even period might appear as an intermediate phase between the 1/2 plateau and the 1/3 SSP. One possible scenario to explain the experimental results is that a 1/2 SSP appears at $H_{c6}$ just above the 1/2 plateau. In this case, the translational symmetry is the same for the 1/2 phases, which may lead to similar values for $\lambda_1 E' + \lambda_2 E''$. Thus, the ultrasound technique might be less sensitive to detect the transition from the 1/2 plateau to a 1/2 SSP. In contrast, the transition from a 1/2 SSP to a 1/3 SSP would be clearly detected because the magnetic unit cell drastically changes. The anomaly at $H_{c7}$ might correspond to this phase transition.

From the theoretical point of view, one can identify a metastable 1/2 SSP, but it is higher in energy than the 1/3 SSP for $J'/J \sim 0.63$. Therefore, the simple Shastry-Sutherland model does not explain the proposed scenario. To stabilize the 1/2 SSP, some additional terms need to be included in Eq. (1). One magnetic interaction neglected in the iPEPS is the interlayer coupling[34,35]. Since the interlayer coupling is AFM, it is energetically not favorable to have two dimers of aligned spins on top of each other. Thanks to the checkerboard structure of the 1/2 plateau and 1/2 SSP, it is possible to always have a polarized dimer on top of a singlet dimer in 3D, leading to lower interlayer energy compared to that of the 1/3 SSP. The same holds true for the 1/2 SSP with the checkerboard-like structure, indicating that the 1/2 SSP could be stabilized with this term. Since the method used in the present work, iPEPS, is a purely two-dimensional approach, the inclusion of the interlayer coupling is left for future work. Another important magnetic interaction is the Dzyaloshinskii-Moriya (DM) term, which stabilizes canted spin orientations like in SSPs. However, this term would stabilize both the 1/2 SSP and the 1/3 SSP similarly. Thus, the DM term would not change the relative stability of these phases. Finally, the last term which might be relevant for the 1/2 plateau and SSP is the spin-lattice coupling term[36]. In these phases, the checkerboard pattern of triplets leads to a tetragonal-orthorhombic instability of the lattice. The orthorhombic distortion alternatively modulates the intra-dimer coupling $J$, stabilizing the magnetic energy at the expense of the elastic energy. This energy gain might be sufficiently large to stabilize the 1/2 SSP compared to the 1/3 SSP. Further work is needed to fully understand how to improve the agreement between experiment and theory in this field range.

Beyond SCBO, the very nice agreement between the theoretical predictions and the experiments regarding the saturation field and other critical fields demonstrates the power of the experimental approach taken in the present investigation and establishes the combination of ultrasound and magnetostriction measurements with pulsed fields up to 150 T as a rather unique source of information in a field range that has been little explored so far. This opens very interesting perspectives for the high-field study of other quantum magnets,

and more generally of strongly correlated materials with exotic magnetic properties.

## Methods

### Experimental method
The pulsed magnetic fields up to 150 T were generated with help of the vertical single-turn-coil system (STC) in the ISSP, University of Tokyo[37]. We used a liquid $^4$He bath cryostat to keep the sample at 3.2–4.2 K. We note that the sample temperature can change due to the magnetocaloric effect during the pulsed field ($\sim6\,\mu s$)[17,29]. High-quality single crystals of SCBO were grown by a traveling solvent floating zone method[38]. We used the one ($2 \times 1 \times 1\,mm^3$) for the ultrasound and another one ($2 \times 1 \times 0.3\,mm^3$) for the magnetostriction measurement. Magnetic fields were always applied along the $c$ axis in our experiments.

We performed the ultrasound measurements by using the continuous-wave excitation technique[39]. Ultrasound waves with the frequency of 20–40 MHz were excited by a LiNbO$_3$ resonance transducer attached to the surface of the crystal. The transmitted waves were detected by another transducer and recorded by a digital oscilloscope. The recorded signals were analyzed by using the numerical lock-in technique, and the phase change was converted to the relative change of the sound velocity $\Delta v/v_0$. With this technique, one can obtain reliable results around the peak of the pulsed field where the field sweep rate slows down. Therefore, we repeated the measurements with different peak fields and extracted the reproducible part of the results. We also performed the ultrasound measurements up to 83 T by using the ultrasound pulse-echo technique with a dual-pulse magnet in the HLD, Dresden[40]. We measured two in-plane modes, $c_{66}$ (**k**||[100], **u**||[010]) and $c_{11}$ (**k**||**u**||[100]), where **k** (**u**) is the propagation (displacement) vector.

We performed the magnetostriction experiments using the Fiber-Bragg grating (FBG) fixed onto the crystal and the optical filter method[41]. When the sample length changed, the Bragg wavelength of the reflected light also changed. By using a band-pass filter with a band edge close to the Bragg wavelength, the reflection wavelength shift was detected as an amplitude change. This scheme allows us to measure magnetostriction at high frequency of 100 MHz. We used an amplified spontaneous emission source as an incident broadband near-infrared light source. In this study, we fixed the FBG using the low-temperature glue SK-229 to detect the longitudinal magnetostriction along the $c$ axis, $L_c$. In addition, the fiber and sample were put inside a vacuum grease to attenuate the sample vibration caused by the magneto-structural phase transitions[42,43]. Because of the large amount of grease coupled to the FBG, the detected magnetostriction was reduced to ~20% of the reported value[17]. Nevertheless, the obtained magnetostriction was qualitatively reproducible and reflected the magnetic-field-induced phase transitions.

### Theoretical method
We have used iPEPS to map out the phase diagram at high magnetic fields, complementing previous results at low[44] and intermediate fields[3,12]. An iPEPS is a variational tensor network ansatz to represent 2D ground states in the thermodynamic limit[31–33] and can be seen as a higher-dimensional generalization of matrix product states. The ansatz consists of a unit cell of tensors which is periodically repeated on the infinite 2D lattice, with one tensor per dimer. We have tested various unit cell sizes to identify the relevant magnetic structures, including rectangular unit cells up to $10 \times 10$, diagonal stripe unit cells up to periods 18, and various non-rectangular unit cells[44]. The accuracy of the ansatz is systematically controlled by the bond dimension $D$ of the tensors. The optimization of the variational parameters is done based on an imaginary time evolution using a simple update[45] and cluster update[46] approach, which provides good estimates of ground state energies while being computationally affordable even for very large

unit cell sizes. The approximate contraction of the 2D tensor network is done by a variant[47,48] of the corner-transfer matrix method[49,50], where the contraction dimension $\chi$ is kept large enough so that contraction errors are negligible. For more details on the iPEPS approach, we refer to Refs. [51–53].

## Data availability

All data needed to evaluate the conclusions in the paper are present in the paper and/or the Supplementary Information. Additional data related to this paper may be requested from the authors.

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

## Acknowledgements

We acknowledge the support of the HLD at HZDR, member of the European Magnetic Field Laboratory (EMFL), and the Deutsche Forschungsgemeinschaft (DFG) under SFB 1143. This work was partly supported by JSPS KAKENHI, Grant-in-Aid for Scientific Research (Nos. JP20K14403, JP22H00104, JP23H04859, and JP23H04861) and JSPS Bilateral Joint Research Projects (JPJSBP120193507), by the European Research Council (ERC) under the European Union's Horizon 2020 research and innovation programme (grant agreement Nos. 677061 and 101001604), and by the Swiss National Science Foundation (grant No.

212082). The crystal structure figures have been created by using the visualization software VESTA[54].

## Author contributions

T.N. and S.Z. designed and initiated the project. C.Z. and H.K. grew the single crystals. T.N., A.M., and S.Z. performed the ultrasound experiments. T.N., Y.I., and A.I. performed the magnetostriction experiments. P.C. and F.M. performed the iPEPS calculations. All authors discussed the results. T.N. and P.C. prepared the manuscript under the supervision of S.Z., Y.K., Y.H.M., and F.M.

## Competing interests

The authors declare no competing interests.
