## [Peer Review File · Nature Communications]

REVIEWER COMMENTS

Reviewer #1 (Remarks to the Author):

In this manuscript, the authors have used ultrasound and magnetostriction techniques combined with extensive tensor network simulations to study the Shastry-Sutherland compound $\text{SrCu}_2(\text{BO}_3)_2$ up to the saturation magnetic field of 140T. Several spin-supersolid phases (SSP) are revealed by the iPEPS simulation and confirmed by the experiment.

In my opinion, this is an interesting and important work that combines both experimental efforts and iPEPS based numerical methods to reveal new phases in a quantum material. Overall, both approaches agree well with each other, both in the critical field and the saturation field. The discrepancy between the two approaches, e.g., the period-10 SSP is not found by the experiment, are also carefully explained. Therefore, this manuscript could be considered to publish in Nature Communications.

However, before publishing on Nature Communications, I have some suggestions and comments for the authors to consider.

1) In the phase diagram revealed by iPEPS (Fig.3), there are some narrow regions corresponding to phases not detected by the experiments, e.g., period-10 SSP. In the iPEPS simulations, the authors have used simple update and cluster update in the iPEPS optimization, both of which are known to have insufficient precision in highly competing region. Could this small region be due to the local minima in iPEPS simulation?

2) In the discussion part, the authors mentioned that ultrasound and magnetostriction measurement with high fields could provide unique information for quantum materials. In some quantum magnets, one would care more about the phases without external magnetic field, e.g. some quantum spin liquid candidates. Could the authors discuss the information one could get from the ultrasound and magnetostriction measurements in these kind of systems?

Ji-Yao Chen

Reviewer #2 (Remarks to the Author):

The authors present sound velocity attenuation and dilation measurements in $\text{SrCu}_2(\text{BO}_3)_2$, carried out in pulsed magnetic fields to 150T. The experimental data is compared against computations for the (H,T) phase diagram using tensor-network calculations (iPEPS). The main findings include the observation of unexpected significant sound velocity attenuation (Dv/v_0) in the $M=\frac{1}{2}M_s$ magnetization state, the interpretation of an anomaly in Dv/v_0 at $H=140\text{T}$ as the field-induced saturated magnetization state, and the interpretation of smaller anomalies in the experimental data as transitions between predicted spin super solid phases (SSP) of different periodicity. The manuscript is well written, but I have some concerns on the experimental side that need clarification.

All the experimental results are presented in Fig.2, with the sound attenuation vs field displayed in panel (a) and magnetostriction vs field in panel (b). Data in panel (c) is not new but a reproduction of data published earlier by some co-authors. The magnetostriction data DL/L vs field seem to support the observation of the end of the $M = \frac{1}{2} M_s$ plateau at $H_{c6} = 108\text{T}$ by earlier magnetization vs field data (panel (c)). Yet, it is virtually impossible to identify unambiguously any additional anomalies. Considering the noise level and the lack of reproducibility of some bigger features at the approx. location of 80T (see supp material) I do not see the justification for the labeling of H_{c8} and H_{c9} in Fig 2b.

The Dv/v_0 data for the c66 acoustic mode in panel (a) is clearer. It is worrisome, however, that some anomalies are only observed in the transition between data of different colors, and one must assign these to transitions cautiously. With the above caveats, the sound velocity attenuation for the c66 mode is the cleanest. I wonder, can the discrepancy between the magnetostriction determination of $H_{c6} = 108\text{T}-116\text{T}$ (or, possibly, $112\text{T} \pm 4\text{T}$) and the sudden change in Dv/v_0 at 116T be explained by the time delay between the magnetic field and ultrasound propagation across the sample?

The sound velocity data taken in the non-destructive magnet in $\text{SrCu}_2(\text{BO}_3)_2$ reveal an anomaly at magnetic fields consistent with the previously observed $2/5$ plateau. The data in the STC magnet show an unexpectedly strong attenuation at the previously observed $M = \frac{1}{2} M_s$ plateau, a clear end of this plateau, and, potentially, an anomaly that agrees with the theoretical prediction for full saturation of the magnetization. However, the anomaly at " H_{c9} " is observed in only one of the traces, and it isn't clear if it happens during field up-sweep or down-sweep. Is the anomaly seen during magnetic field up-sweep (sample likely colder) or down-sweep (sample likely hotter)? If seen during the magnetic field down-sweep, how is this interpreted?

Concluding, the manuscript is very interesting in exploring the physics of a highly frustrated spin system in a very difficult-to-reach range of experimental parameters and providing an accompanying theoretical framework. The spin super solid phases are intriguing, and very likely the data presented by Nomura et al., will trigger further research. I, however, cannot recommend publication in Nature Communications in the current version.

Reviewer #3 (Remarks to the Author):

The authors report results on magnetic properties of the quantum magnet $\text{SrCu}_2(\text{BO}_3)_2$ at high magnetic fields. In particular they reach ~ 150 T while previous measurements with other techniques reached 118T. They measure the variations of the speed of sound and magnetostriction to detect phase transitions as a function of applied magnetic field. The nature of the phases is inferred by iPEPs calculations and previous knowledge about this material.

From the experimental point of view, those experiments are very challenging. The results are interesting and measuring SCBO magnetic properties up to saturation is of particular interest, in principle worth publication, provided the authors are able to give important clarifications to better substantiate some of their claims as explained below.

Three main types of experimental results are shown 1) the change of the speed of sound in the c11 mode. 2) change of the speed of sound in the c66 mode and 3) the magnetostriction which measures the change in the length of the sample in the c-axis direction.

The c11 mode shows almost no variation with magnetic field.

The c66 mode shows strong variations.

The magnetostriction shows some variations, but in a very noisy environment and are overall less convincing.

Starting with the magnetostriction data presented in Fig 2b.

1) There are two orange and two blue lines. Do they correspond to field up and field down? If so it should be indicated.

2) The position and existence of the anomalies labeled Hc6 to Hc9 does not seem strongly consistent among the four lines presented. At best there is an increase around Hc6 that saturates around Hc7 or somehow higher.

3) There are many other bumps at lower field value and it is not very clear how the Hc6 to Hc9 could be detected from the magnetostriction data alone.

4) Can this magnetostriction data be used to detect the phase changes below Hc6 (~100T)? The naming suggests there are Hc1 to Hc5, are those anyhow noticeable in the presented lines of Fig 2b?

For the c66 speed of sound data

1) Same comment about the multiple lines of a single color. Are those field up and down or different runs?

2) There seems to be a good match between the variations in this mode and the lower field features of the SCBO phase diagram.

3) The Hc7 transition is clear.

4) The Hc8 is described as detected by a change of slope, this is not obvious while looking at the data. The green and blue lines in Fig 2a could as well be a one smooth feature.

5) The Hc9 is visible in only one of the two lines shown in Fig 2a. Why is not in the other line?

I think the authors should make it clearer how they detect and assign the critical fields from the presented data and probably add this in a supplementary file.

They should also make it clearer as to which extent those critical field can be objectively inferred by each method individually or how much they rely on the confirmation from the other method or from previous magnetization measurements.

Other comments:

Abstract mentions 140T while the text mentions 150T

Abbreviations like FGB and STC are not defined in the main text.

The formatting does not make it very clear if the last paragraph is related to section IV or is an independent 'conclusion'.

Reply to Reviewer #1: Ji-Yao Chen

In this manuscript, the authors have used ultrasound and magnetostriction techniques combined with extensive tensor network simulations to study the Shastry-Sutherland compound $\text{SrCu}_2(\text{BO}_3)_2$ up to the saturation magnetic field of 140T. Several spin-supersolid phases (SSP) are revealed by the iPEPS simulation and confirmed by the experiment.

In my opinion, this is an interesting and important work that combines both experimental efforts and iPEPS based numerical methods to reveal new phases in a quantum material. Overall, both approaches agree well with each other, both in the critical field and the saturation field. The discrepancy between the two approaches, e.g., the period-10 SSP is not found by the experiment, are also carefully explained. Therefore, this manuscript could be considered to publish in Nature Communications.

We appreciate your careful readings and positive comments on our manuscript. We believe that our combined experimental and theoretical study has opened a new horizon of this research field. The followings are our replies to your comments/questions.

1) In the phase diagram revealed by iPEPS (Fig.3), there are some narrow regions corresponding to phases not detected by the experiments, e.g., period-10 SSP. In the iPEPS simulations, the authors have used simple update and cluster update in the iPEPS optimization, both of which are known to have insufficient precision in highly competing region. Could this small region be due to the local minima in iPEPS simulation?

The period-10 SSP is indeed extremely narrow, particularly at the relevant value of $J'/J=0.63$, where its extent is hardly visible anymore in the phase diagram in Fig. 3. However, its extent increases with decreasing J'/J , and in order to confirm this trend, we have performed additional simulations down to $J'/J=0.56$ at which the extent is already 6 times as large as for $J'/J=0.63$, and the energy difference between the competing states roughly an order of magnitude larger. Thus, this phase is definitely relevant for the phase diagram of the Shastry-Sutherland model, but it coincidentally terminates around $J'/J=0.63$.

Regarding the approach, it is true that the full update optimization or energy minimization would yield a higher accuracy for a given bond dimension D , however, these approaches are limited to substantially smaller D (in particular due to the large cell sizes and absence of $U(1)$ symmetries), such that the cluster update, which can be pushed to large D , offers the best possible approach here.

In the revised manuscript, we have added a comment to mention these additional results regarding the increasing extent of the period-10 SSP with decreasing J'/J , and we have also included this new data in the supplemental material.

2) In the discussion part, the authors mentioned that ultrasound and magnetostriction measurement with high fields could provide unique information for quantum materials. In some quantum magnets, one would care more about the phases without external magnetic field, e.g. some quantum spin liquid candidates. Could the authors discuss the information one could get from the ultrasound and magnetostriction measurements in these kind of systems?

The ultrasound and magnetostriction techniques are particularly sensitive to the phase transitions breaking symmetries inherent to the crystal structure. For the case of magneto-structural transitions, the active elastic mode shows extremely large response as observed in our results [c66 mode in the 1/2 plateau, Fig. 2a].

For the case of quantum spin liquids, the discussion is not trivial because this exotic state does not really break the crystallographic symmetries. There are several theoretical and experimental studies proposing ultrasound experiments on the quantum-spin-liquid candidates where spin-phonon interaction might result in characteristic elastic properties.

[Y. Zhou and P. A. Lee, Phys. Rev. Lett. 106, 056402 (2011).]

[K. Feng, A. Shiralieva, and N. B. Perkins, Phys. Rev. B 106, 144424 (2022).]

For experimental results, see

[M. Poirier, M.-O. Proulx, and R. Kato, Phys. Rev. B 90, 045147 (2014).]

[M. Poirier, M. de Lafontaine, K. Miyagawa, K. Kanoda, and Y. Shimizu, Phys. Rev. B 89, 045138 (2014).]

[Nan Tang et al., Nat. Phys. **19**, 92 (2023)]

[A. Hauspurg et al., <http://arxiv.org/abs/2303.09288>]

We also believe that elastic properties (elastic constants and thermal expansion coefficients) are a powerful probe for studying the quantum-spin liquids, although the number of research for this direction has been limited so far. We are also considering these topics for our further research. Thank you for your suggestion.

Reply to Reviewer #2

The authors present sound velocity attenuation and dilation measurements in $\text{SrCu}_2(\text{BO}_3)_2$, carried out in pulsed magnetic fields to 150T. The experimental data is compared against computations for the (H,T) phase diagram using tensor-network calculations (iPEPS). The main findings include the observation of unexpected significant sound velocity attenuation (Dv/v_0) in the $M=1/2M_s$ magnetization state, the interpretation of an anomaly in Dv/v_0 at $H=140\text{T}$ as the field-induced saturated magnetization state, and the interpretation of smaller anomalies in the experimental data as transitions between predicted spin super solid phases (SSP) of different periodicity. The manuscript is well written, but I have some concerns on the experimental side that need clarification.

We appreciate your careful reading and positive comments on our manuscript. The followings are our replies to your comments/questions.

All the experimental results are presented in Fig.2, with the sound attenuation vs field displayed in panel (a) and magnetostriction vs field in panel (b). Data in panel (c) is not new but a reproduction of data published earlier by some co-authors. The magnetostriction data DL/L vs field seem to support the observation of the end of the $M = 1/2 M_s$ plateau at $H_{c6} = 108\text{T}$ by earlier magnetization vs field data (panel (c)). Yet, it is virtually impossible to identify unambiguously any additional anomalies. Considering the noise level and the lack of reproducibility of some bigger features at the approx. location of 80T (see supp material) I do not see the justification for the labeling of H_{c8} and H_{c9} in Fig 2b.

Figure R1 shows the field up- and down-sweeps of the magnetostriction curves with the guide dashed lines to define the phase boundary (data are reproduced from the main text). Although the noise level is high, we can locate H_{c8} and H_{c9} both in the up- and down-sweeps. The relatively weak features in the down-sweep might be due to the temperature increase and/or the slow dynamics of lattice. The relatively worse reproducibility below 80 T (H_{c1} - H_{c5}) is partly because of the higher dB/dt in this magnetic field range.

Fig. R1: Magnetostriction curves for the field up- (solid) and down- (dotted) sweeps.
Dashed lines are guide to the eye.

The Dv/v_0 data for the c66 acoustic mode in panel (a) is clearer. It is worrisome, however, that some anomalies are only observed in the transition between data of different colors, and one must assign these to transitions cautiously. With the above caveats, the sound velocity attenuation for the c66 mode is the cleanest. I wonder, can the discrepancy between the magnetostriction determination of $H_{c6} = 108\text{T}-116\text{T}$ (or, possibly, $112\text{T} \pm 4\text{T}$) and the sudden change in Dv/v_0 at 116T be explained by the time delay between the magnetic field and ultrasound propagation across the sample?

We agree that the ultrasound results (c66 mode) are the most reliable data to determine the phase boundaries. The end of the $1/2$ plateau is observed at the different critical fields depending on the experimental techniques ($\sim 108\text{ T}$ with magnetization and $\sim 116\text{ T}$ with ultrasound). Therefore, we have repeated the ultrasound experiment more than 10 times with different maximum fields. Figure R2 shows the data collection with the maximum fields of $100\text{-}120\text{ T}$ at 4.2 K (see also the data in the main text and the supplementary obtained at 3.2 K). We note that the sweep rate slows down near the top of the pulsed

magnetic field and the propagation time of the ultrasound (~ 300 ns) becomes negligible around the peak field. These data collections clearly show that no anomaly is observed when the peak field is lower than 115 T. Therefore, we can conclude that the different critical fields ($H_{c6}=108$ T, $H_{c7}=116$ T) are most likely not because of the delay of the ultrasound propagation.

Fig. R2: Phase shift of the acoustic wave for the c66 mode versus magnetic field at 4.2 K.

The green and blue bands indicate that the phase change is too fast to unwrap it reliably.

The orange dotted line indicate H_{c6} where the magnetization and magnetostriction show anomaly.

The sound velocity data taken in the non-destructive magnet in $\text{SrCu}_2(\text{BO}_3)_2$ reveal an anomaly at magnetic fields consistent with the previously observed $2/5$ plateau. The data in the STC magnet show an unexpectedly strong attenuation at the previously observed $M = \frac{1}{2} M_s$ plateau, a clear end of this plateau, and, potentially, an anomaly that agrees with the theoretical prediction for full saturation of the magnetization. However, the anomaly at “ H_{c9} ” is observed in only one of the traces, and it isn’t clear if it happens during field up-sweep or down-sweep. Is the anomaly seen during magnetic field up-sweep (sample likely colder) or down-sweep (sample likely hotter)? If seen during the magnetic field down-sweep, how is this interpreted?

In the revised manuscript, we show the field down-sweep with the dotted line. The anomaly at H_{c9} is observed both in the field-up and -down sweeps, but the more pronounced feature appears in the up sweep. As the referee suggests, one reason might be the temperature change (magnetocaloric effect and/or eddy-current heating of the wires nearby). Another possibility is the lattice dynamics, which might take \sim microsecond to relax the structure. Temperature change during the pulsed magnetic field is a long-standing issue in the high-field community, and the reliable technique to detect it has not yet been developed for the single-turn coil. In the main text, we add a comment:

“The slight difference between the field up- and down-sweeps might be due to the temperature change and/or the slow lattice dynamics at the magneto-structural transitions.”

Concluding, the manuscript is very interesting in exploring the physics of a highly frustrated spin system in a very difficult-to-reach range of experimental parameters and providing an accompanying theoretical framework. The spin super solid phases are intriguing, and very likely the data presented by Nomura et al., will trigger further research. I, however, cannot recommend publication in Nature Communications in the current version.

We also believe that our combined experimental and theoretical study will trigger new research on quantum-spin and correlated-electron systems with geometrical frustration. We hope that our revised manuscript has removed all your concerns and meets the criteria of Nature Communications.

Reply to Reviewer #3

The authors report results on magnetic properties of the quantum magnet $\text{SrCu}_2(\text{BO}_3)_2$ at high magnetic fields. In particular they reach ~ 150 T while previous measurements with other techniques reached 118T. They measure the variations of the speed of sound and magnetostriction to detect phase transitions as a function of applied magnetic field. The nature of the phases is inferred by iPEPs calculations and previous knowledge about this material.

From the experimental point of view, those experiments are very challenging. The results are interesting and measuring SCBO magnetic properties up to saturation is of particular interest, in principle worth publication, provided the authors are able to give important clarifications to better substantiate some of their claims as explained below.

We appreciate your careful reading and positive comments on our manuscript. The followings are our replies to the comments/questions.

Three main types of experimental results are shown 1) the change of the speed of sound in the c_{11} mode. 2) change of the speed of sound in the c_{66} mode and 3) the magnetostriction which measures the change in the length of the sample in the c -axis direction. The c_{11} mode shows almost no variation with magnetic field. The c_{66} mode shows strong variations. The magnetostriction shows some variations, but in a very noisy environment and are overall less convincing.

Starting with the magnetostriction data presented in Fig 2b.

1) There are two orange and two blue lines. Do they correspond to field up and field down? If so it should be indicated.

In the revised manuscript, we show the results for the field up- (solid line) and down-sweeps (dotted line). Thank you for the comment.

2) The position and existence of the anomalies labeled H_{c6} to H_{c9} does not seem strongly consistent among the four lines presented. At best there is an increase around H_{c6} that saturates around H_{c7} or somehow higher.

As the referee pointed out, the four lines of magnetostriction (field up- and down-sweeps) show the increase at H_{c6} and saturation at H_{c7} in Fig. 2b. For quantitative value definitions of the critical fields, please see our reply to your next comment Q/A(3).

3) They are many other bumps at lower field value and it is not very clear how the Hc6 to Hc9 could be detected from the magnetostriction data alone.

Figure R1 shows the field up- and down-sweeps of the magnetostriction curves with the guide lines to define the phase boundary (data are reproduced from the main text). Although the noise level is high, we can locate Hc6, Hc7, Hc8, and Hc9. The relatively weak features in the down-sweep might be due to the temperature increase and/or the slow dynamics of lattice. The relatively worse reproducibility below 80 T (Hc1-Hc5) is partly because of a higher dB/dt value in this field range.

Fig. R1: Magnetostriction curves for the field up- (solid) and down- (dotted) sweeps. Dashed lines are guide to the eye.

4) Can this magnetostriction data be used to detect the phase changes below Hc6 (~100T)? The naming suggests there are Hc1 to Hc5, are those anyhow noticeable in the presented lines of Fig 2b?

Because of the microsecond duration of the STC technique and very high sweep rate around Hc1-

Hc5 (50 MT/s), the fine detection of these anomalies is extremely challenging. As shown in Fig. R1, (Hc1-Hc2) and Hc3 are detected with our S/N ratio. With smaller pulsed fields (and smaller charging voltage), the S/N ratio around Hc1-Hc5 would be improved. Note, that the observed tendency for the magnetostriction increase between 20 and 50 T and the magnetostriction plateau above 50 T is compatible with the results obtained in 100 T non-destructive pulsed magnet [Jaime2012].

For the c66 speed of sound data

1) Same comment about the multiple lines of a single colors. Are those field up and down or different runs?

In the revised manuscript, we show the field up- and down-sweeps by different lines.

2) There seems to be a good match between the variations in this mode and the lower field features of the SCBO phase diagram.

Indeed, this acoustic mode (c66 mode) seems quite sensitive for all the phase boundaries. The reasons why the c66 mode is more sensitive than the c11 mode are discussed in the main text.

3) The Hc7 transition is clear.

Yes, it is clear and discontinuous, indicating the first order transition.

4) The Hc8 is described as detected by a change of slope, this is not obvious while looking at the data. The green and blue lines in Fig 2a could as well be a one smooth feature.

Figure R3 shows the ultrasound data with guide lines to determine the phase boundaries. Indeed, the anomaly Hc8 corresponds to the change of the slope. The slope change looks rather discontinuous at the crossing point of the guide lines (Hc8). The magnetostriction data (Fig. R1) also support this phase boundary, and it can be consistently explained by the iPEPS calculations.

Fig. R3: Ultrasound data with the guide lines of the phase boundaries.

5) The H_{c9} is visible in only one of the two lines shown in Fig 2a. Why is not in the other line?

The anomaly at H_{c9} is visible for both the field-up and -down sweeps, although the sharpness of the anomaly is quite different. It might be related to the lattice dynamics and/or the temperature change during the pulse, however, the resolution of our data is not sufficient to discuss the detailed mechanism of the difference.

I think the authors should make it clearer how they detect and assign the critical fields from the presented data and probably add this in a supplementary file.

We have added Figs. R1 and R3 with a new section in the supplementary information. Thank you for the advice.

They should also make it clearer as to which extent those critical field can be objectively inferred by each method individually or how much they rely on the confirmation from the other method or from previous magnetization measurements.

As the referee pointed out, the most reliable information on the phase boundary is obtained by the ultrasound results (c66 mode). Based on these results, Hc7, Hc8, and Hc9 are determined. Magnetostriction data are mainly used to support the reliability of the ultrasound data. The critical fields and error ranges of these phase boundaries are summarized in Table I. These values are independently estimated although Hc8 is carefully assigned based on the ultrasound and magnetostriction data. We added these discussions in the supplementary information.

Other comments:

Abstract mentions 140T while the text mentions 150T

We meant that we performed the experiment up to 150 T and observed the saturation field of 140 T. To avoid the misunderstanding we added in the abstract "... of 140 T and beyond".

Abbreviations like FGB and STC are not defined in the main text.

We have added these definitions. Thank you for the careful reading.

The formatting does not make it very clear if the last paragraph is related to section IV or is an independent 'conclusion'.

We followed the Nature Physics guidelines "Concluding paragraphs that do no more than summarize the conclusions presented elsewhere in the manuscript are not permitted". In this sense, the last paragraph belongs to the section IV, though it contains some summarizing remarks.

List of revisions (highlighted by red in the text)

[1] Abstract: modify the sentence, “140 T and beyond”

[2] Figure 1 updated: the field-down sweep data are shown by the dotted lines

[3] Figure 1 caption: add a sentence

“The results in the field up- and down-sweeps are shown by the solid and dotted lines, respectively”

[4] Sec. I, 2nd paragraph: add a sentence

“The slight difference between the field up- and down-sweeps might be due to the temperature change and/or the lattice dynamics at the magneto-structural transitions.”

[5] Sec. II, 2nd paragraph: add a sentence

"We confirmed that the extent of the period-10 SSP further increases with decreasing J'/J from additional simulations down to $J'/J=0.56$."

[6] Supplementary: add a new section II “Critical fields” with Figs. S6 and S7

[7] Supplementary: add a new section III “Extent of the period-10 supersolid phase” with Fig. S8

Minor revisions

[8] add the definitions of STC and FBG

[9] add the email address, nomura.toshihiro95@gmail.com

REVIEWERS' COMMENTS

Reviewer #1 (Remarks to the Author):

In the reply and revised manuscript, the authors have considered my comments with a satisfactory reply. I therefore recommend publication in Nature Communications.

-- Ji-Yao Chen

Reviewer #2 (Remarks to the Author):

Manuscript: Unveiling new quantum phases in the Shastry-Sutherland compound $\text{SrCu}_2(\text{BO}_3)_2$ up to the saturation magnetic field by T. Nomura et al.

The authors have addressed my concerns satisfactorily, and I hence recommend publication of the revised version in Nature Communications.

Reviewer #3 (Remarks to the Author):

The authors have improved the manuscript and answered most questions by the reviewers.

As reviewer # 2 also pointed the determination of H_{c8} and H_{c9} is not obvious. The authors tried to better explain how they detect those anomalies. While there are signal variations at the locations indicated they are mostly weak compared to the noise present in the data. For example, the best hint I can find for H_{c9} is the field up scan of $c66$, while it is hardly detectable in the field down. Combined to the fact that H_{c7} does not appear in the iPEPS calculation, three of the critical fields have some issues.

That said, these experiments are very challenging and are not easy to repeat, hence their publication can be useful for the community.

The combined consideration of magnetostriction, ultrasound and iPEPS gives some strength to the conclusion of the authors. The experimental results and the theoretical results are interesting on their own and show overlap.

Therefore, in my opinion, the quality of the detection of two of the four presented experimental transitions is weak but the overall interest of the manuscript remains strong.